# Safety Profile of Ambulatory Prostatic Artery Embolization after a Significant Learning Curve: Update on Adverse Events

**DOI:** 10.3390/jpm12081261

**Published:** 2022-07-31

**Authors:** Gregory Amouyal, Louis Tournier, Constance De Margerie-Mellon, Atanas Pachev, Jessica Assouline, Damien Bouda, Cédric De Bazelaire, Florent Marques, Solenne Le Strat, François Desgrandchamps, Eric De Kerviler

**Affiliations:** 1Ramsay Santé—Hôpital Privé Geoffroy Saint-Hilaire, 75005 Paris, France; florent.marques@gmail.com (F.M.); docteurlestrat@gmail.com (S.L.S.); 2Radiology Department, Hôpital Saint-Louis, 75010 Paris, France; ltourn22@gmail.com (L.T.); constance.de-margerie@aphp.fr (C.D.M.-M.); atanas.pachev@aphp.fr (A.P.); jessica.assouline@aphp.fr (J.A.); damien.bouda@aphp.fr (D.B.); cedric.de-bazelaire@aphp.fr (C.D.B.); eric.de-kerviler@aphp.fr (E.D.K.); 3Faculté de Médecine, Université Paris Cité, 75006 Paris, France; francois.desgrandchamps@aphp.fr; 4Urology Department, Hôpital Saint-Louis, 75010 Paris, France; 5SRHI/CEA—Institut de Recherche Clinique Saint-Louis, Hôpital Saint-Louis, 75010 Paris, France

**Keywords:** prostatic hyperplasia, embolization, therapeutic, endovascular procedure, radiology, interventional, prostate

## Abstract

Background: to report the safety of outpatient prostatic artery embolization (PAE) after a significant learning curve. Methods: a retrospective bi-institutional study was conducted between June 2018 and April 2022 on 311 consecutive patients, with a mean age of 69 years ± 9.8 (47–102), treated by outpatient PAE. Indications included lower urinary tract symptoms, acute urinary retention, and hematuria. When needed, 3D-imaging and/or coil protection of extra-prostatic supplies were performed to avoid non-target embolization. Adverse events were monitored at 1-, 6-, and 12-month follow-ups. Results: bilateral PAE was achieved in 305/311 (98.1%). Mean dose area product/fluoroscopy times were 16,408.3 ± 12,078.9 (2959–81,608) μGy.m^2^/36.3 ± 1.7 (11–97) minutes. Coil protection was performed on 67/311 (21.5%) patients in 78 vesical, penile, or rectal supplies. Embolization-related adverse events varied between 0 and 2.6%, access-site adverse events between 0 and 18%, and were all minor. There was no major event. Conclusion: outpatient PAE performed after achieving a significant learning curve may lead to a decreased and low rate of adverse events. Experience in arterial anatomy and coil protection may play a role in safety, but the necessity of the latter in some patterns may need confirmation by additional studies in randomized designs.

## 1. Introduction

For about ten years, prostatic artery embolization (PAE) has been described as a novel mini-invasive procedure and alternative treatment to surgery for symptomatic benign prostatic hyperplasia (BPH) [1,2]. To date, the literature has shown the efficacy of PAE on lower urinary tract symptoms (LUTSs) close to similar to surgical options, and the majority reports fewer minor events [3,4,5,6], its safety profile being mainly based on reports assessing PAE from 2011 to 2016 [7,8,9,10].

New evidence has since been published on anatomy [11,12,13] and technical achievements, such as the coil/gelatin protection of extra-prostatic supplies during PAE [14,15], and the use of different embolic types and sizes [16,17] or new devices, such as a balloon occlusion micro catheter [18,19,20,21], all aiming to improve efficacy and decrease non-target embolization. Obviously, in addition to these new tools/technical evolutions, another major asset for safety in performing PAE is the learning curve.

No recent experience of the overall adverse events has been reported since the improvement of knowledge and techniques. In the present study, the results of short-term complications following PAE performed after a significant learning curve are assessed.

## 2. Materials and Methods

This bi-institutional retrospective study was performed on 311 consecutive male patients between June 2018 and April 2022, with a mean age of 69 years ± 9.8 (47–102). Indications for PAE were patients manifesting symptomatic BPH, described as prostatic volume >35 mL associated with moderate to severe LUTSs, defined as an international prostatic symptoms score (IPSS) > 8 or a quality-of-life score (QoL) > 3/7, and showing failure of optimal medical treatment, or patients with acute urinary retention (AUR) due to BPH and failure of trial without an outer catheter after at least 48 h of alpha-blocker medication, or BPH leading to repeated episodes of bothersome macroscopic hematuria. The exclusion criteria were prostate or bladder cancer, urethral stricture, complicated BPH leading to the dilatation of urinary cavities, and factors preventing the performance of PAE, such as the occlusion of any iliac artery or advanced dementia.

Indications were validated in the clinic by a urologist and interventional radiologist.

Institutional Review Board consent was obtained from each patient for this study. The baseline characteristics of the population are presented in Table 1.

### 2.1. PAE Procedure

PAE was performed during ambulatory care for all patients, with the intent to use transradial access (TRA), for comfort issues and in order to shorten their hospital stay. When the left TRA was unfeasible, right transfemoral access (TFA) was used. No Foley catheter was inserted for the performance of the intervention. All patients received per-procedure antibiopropylaxy (intravenous, 1.5 g of cefazoline or 600 mg of clindamycin in case of an allergy to penicillin); no pre- nor post-procedural antibiotherapy was provided. PAE was performed in a 4D CT suite (center 1) or in a c-arm floor-mounted Angio suite equipped with a cone-beam CT (center 2), using 5-Fr TRA or TFA under local anesthesia (center 1) or local anesthesia and intravenous neurolept analgesia (center 2) composed of a mix of intravenous ketamine, midazolam, and fentanyl.

Following selective internal iliac angiography using a 5Fr catheter (Merit Medical, Salt Lake City, UT, USA), the super selective catheterization and angiography of the prostatic artery (PA) was performed on each side using a 2.0-Fr micro catheter (Merit Medical) and 0.014′ micro guide wire (Boston Scientifics, Malborough, MA, USA).

Angio CT or cone-beam CT were performed in selected cases for the detection of the origin of PA, as guidance for selective catheterization, or when angiography alone could not confirm the extra-prostatic supply from the PA.

Extra-prostatic vesical, penile, or rectal supplies were occluded prior to prostatic embolization using coil protection, as previously described [14,15]. PA with penile supplies, previously described as “pattern B PA” [12], were routinely occluded (after angiographic confirmation of at least the unilateral patency of the internal pudendal arteries), except for situations of a reversed flow in the penile arteries oriented toward the apex of the prostate, preventing the anterograde delivery of the embolic agent in the penile arteries during embolization. Rectal supplies from the PA, named “pattern C” [12], were occluded only when selective prostatic catheterization could not prevent early reflux in the rectal artery in pattern C1 and routinely in case of pattern C2 (distal origin of an accessory rectal artery). In cases of an accessory inferior vesical artery (AIVA) or a vesical anastomose originating from the distal part of the PA, coil protection was used only when early reflux was observed during selective prostatic angiography to prevent non-target embolization (Figure 1).

After ruling out or occluding the extra-prostatic supply from the PA, super selective proximal PAE was performed, using 300–500 μm calibrated trisacryl microparticles, until complete stasis. At a complete bilateral PAE, catheters were retrieved and the access site was closed using a hemostatic band for TRA (TR band^®^, Terumo Corporation, Tokyo, Japan, or Prelude Sync^®^, Merit Medical), or a closure device (Exoseal^®^ 5Fr, Cordis, Miami Lakes, FL, USA) in case of TFA. Patients were discharged 75 to 90 min (TRA) or 180 to 240 min (TFA) after the completion of PAE.

Technical success was defined as achieving at least unilateral PAE. Clinical success was defined as an IPPS decrease of at least 8 points, a QoL score decrease of 1 point or score <3, the ability to stop any medication later than 15 days following PAE, the successful retrieval of the Foley catheter at day 15, or the disappearance of hematuria. The removal of the indwelling catheter was delayed by 10 days in case of ongoing urinary tract infection (UTI) after elective oral antibiotic treatment.

Follow-up was performed at 1, 6, and 12 months to assess clinical success using identical documentation as pre-procedural workup. Post-embolization syndrome and minor/major complications following PAE were defined according to the Society of Interventional Radiology classification of complications [22], and were monitored at one month using a standardized questionnaire filled out by the patient. Additional left radial artery Doppler ultrasound simultaneously monitored TRA adverse events. Mid-term adverse events were monitored at a 6-month follow-up, when a short-term adverse event was observed at the 1-month follow-up. Patients were deemed lost to follow-up when the questionnaire regarding adverse events was not filled out at 1 month, or when no documentation regarding an ongoing adverse event or no follow-up exams were obtained at 6- and 12-month visits.

### 2.2. Statistical Analysis

The differences between the baseline and 1-month data were assessed using Student’s paired t-test and R software, version 4.1.1. The results are expressed as mean ± SD (range) and their *p*-values. A *p*-value < 0.05 was considered significant.

### 2.3. Results

A total of 315 patients were referred by the urologist for PAE (Figure 2). Over the course of the pre-procedural assessment, prostatic MRI revealed two cases of advanced prostatic cancer extending to the bladder neck, one case of a T2a staged prostatic cancer, and one case of advanced bladder cancer extending to the prostate. These four patients did not undergo PAE and were referred back to the urologist, because LUTSs were not caused by BPH but by locally advanced cancer (*n* = 3) or because prostatic cancer required oncological treatment (*n* = 1).

A total of 300/311 (96.4%) patients underwent transradial outpatient PAE and 11/311 (3.6%) transfemoral outpatient PAE. Bilateral embolization was performed in 305/311 (98.1%) cases. Mean DAP/fluoroscopy times were 16,408.3 ± 12,078.9 (2959–81,608) μGy.m^2^/36.3 ± 15.7 (11–97) minutes. Angio CT or CBCT was needed for 10/311 (3.2%) patients; 6 mappings of the PA, 4 to rule out extra-prostatic supplies. The procedure characteristics are presented in Table 2 and the distribution of prostatic arterial vasculature in Table 2 and Figure 3 and Figure 4, according to Assis’s and Amouyal’s classification [11,12].

Coil protection was considered as necessary to prevent non-target embolization in 67/311 (21.5%) patients in 78 extra-prostatic supplies. Among pattern B PAs, 30/37 (81%) benefited from coil protection: 4 did not because of a reversed flow in the penile arteries and 3 because the coil protection of pattern B PA had already been performed on the contralateral side. Among C1 and C2 pattern arteries, 20/62 (32%) middle rectal arteries were protected because of early reflux from the prostatic branch (C1) and 5/12 accessory rectal artery (42%) (C2) when its super selective catheterization was achievable. Among AIVA/vesical anastomoses, 23/58 (40%) required protection.

Mean follow-up was 4.35 months.

Post-embolization syndrome was observed in 264/311 (84.9%) patients, and urethral burning and pollakiuria were the predominant symptoms. The mean duration increased for most symptoms with prostate volume (Table 3); the overall mean durations for urethral burning and pollakiuria were 4 ± 3.9 (0–12) and 4.2 ± 3.4 (0–12) days. There were 8/311 (2.6%) cases of transient hematospermia that spontaneously resolved within 1 to 90 days. For each patient, control MRI did not present a seminal vesicle signal or enhancement abnormality. The overall post-embolization symptoms and their durations according to prostate volume are presented in Table 3.

### 2.4. Adverse Events

All adverse events were minor, most occurred in the first 21 days (Table 4 and Table 5), and all but one were monitored at the 1-month follow-up control or reported by the patient prior to the visit. There was no major adverse event. Access-site adverse events are presented in Table 5 and are mostly represented by hematoma.

Embolization-related complications ranged from 0.6 to 2.6%: there were 2/311 (0.6%) cases of urinary tract infection in patients treated for AUR, for whom an indwelling catheter was still in place, occurring at days 10 and 12, manifested by moderate fever and a positive urinary sample test. Both were successfully treated by oral antibiotics for 15 days. Catheter removal was attempted 10 days after relief of fever: one was successful, the other failed, and so did a second attempt 15 days later.

There were 2/311 (0.6%) cases of transient balanitis appearing at days 2 and 3, manifested by bilateral asymptomatic zones of fibrin on the mucosa of the glans that were successfully treated with local antiseptic and antibiotics and resolved within 15 days with no sequelae.

An angiographic review of these two cases reported the presence for each patient of an accessory PA supplying the gland from the apex (“distal pudendal PA”), originating from the distal portion of the internal pudendal artery (IPA), in close proximity to the penile vessels.

There were 8/311 (2.6%) cases of macroscopic hematuria occurring between days 1 and 20, except for one case associated with fragment detachment at day 90. All were spontaneously resolved within 24 h. A retrospective angiographic review revealed a bilateral type-1 origin of the PA according to De Assis (common origin with the vesical arteries) in two patients and a unilateral type-1 origin in six patients. The coil protection of an AIVA was performed on one patient.

There were 2/311 (0.6%) cases of a single episode of rectorrhagia manifested by traces of blood in the stool occurring at days 1 and 3, in a context of constipation, also resolved in 24 h. An angiographic review reported bilateral pattern A origins for both patients and no coil protection of rectal arteries.

There was one case (1/311, 0.3%) of transient worsening of erectile dysfunction following coil protection of a pattern B PA. In this patient, there was a short and proximal occlusion of the ipsilateral IPA, whereas contralateral IPA was patent. As IPA was patent on one side, the decision was made to use coil protection. The revascularization of the occluded IPA was proposed to the patient at day 21 with the aim to improve erectile function, which he refused, and instead preferred tadalafil oral medication with limited efficacy. LUTS improved within 15 days and erectile dysfunction spontaneously improved to reach pre-embolization status within six months with no further need for tadalafil.

There was one case (0.3%) of detachment of the prostatic fragment occurring at three months. At the 1-month follow-up, patient symptoms had improved: IPSS/QoL scores and the prostatic volume changed from 26, 7, and 120 mL to 4, 1, and 80. Approximately three months following PAE, the patient suddenly presented an episode of hematuria associated with a recurrence of dysuria and pelvic pain during urination. He described the spontaneous expulsion of several small fragments of prostatic tissue and a clot during micturition, for 48 h. As the symptoms persisted, the patient visited the urology department and a cystoscopic examination performed five days later revealed a clot and necrotic scar on the median lobe wall, from which a centimetric fragment was removed with no subsequent bleeding. No MRI was performed in this urgent context. Dysuria/pain disappeared after the fragment removal and there was no anejaculation following this partial resection.

There were 5/311 (1.7%) cases of a transient reduction in ejaculate volume lasting from 15 days to 4 months. There were no cases of anejaculation.

There was no case of acute urinary retention, glans, rectal or vesical partial ischemia, or radiodermitis.

Apart from radial artery occlusion at the puncture site, no persistent adverse event was observed at 6- and 12-month follow-up controls.

### 2.5. Clinical Success

The successful removal of the Foley catheter on day 15 for patients treated for AUR was observed in 44/56 (78.6%) patients. Overall clinical success was 268/311 (86.2%) at the 1-month follow-up control: mean IPSS and QoL scores decreased from 18.9 ± 7 (4–35) and 6 ± 1.1 (2–7) to 8.7 ± 7.2 (0–35) (*p* < 0.001) and 3 ± 1.7 (1–7) (*p* < 0.001). There was no significant change in mean IIEF score values (*p* = 0.83); mean maximum urinary flow increased from 8.1 ± 5 (2.4–31) mL/s to 13.4 ± 6.1 (3–32) (*p* < 0.001); and mean post-voiding residue, prostate volume, and total PSA decreased significantly (*p* < 0.001) (Table 6).

## 3. Discussion

The present study of ambulatory PAE reported low rates (0.6 to 2.6%) of embolization-related minor adverse events and no major adverse event. Major adverse events remain rare and each one is estimated to occur in 0.08% to 0.24% of cases [7,8,9,10].

### 3.1. Embolization-Related Adverse Events

As most major and minor complications follow non-target embolization, a possible explanation for their absence/low rate in this report may be the increased knowledge concerning the management of extra-prostatic communications during PAE [14,15] and prostatic arterial anatomy [11,12,13]. Three-dimensional imaging during selective angiography used as a routine practice may help to rule out extra-prostatic supplies, and its need may decrease over time and eventually be restricted to elective cases once a significant learning curve is achieved.

Early minor events ranged from 0.6 to 2.6%, which were lower than those previously reported [7,8,9,10]. Multiple adverse events previously described in the literature and meta-analyses were not observed in this study, which may suggest a significant decrease in non-target embolization and may be explained by the improvement of experience during recent years.

There was no AUR compared to the 4.55, 7, 9 and 7.8% previously described [7,8,9,10]. AUR is favored by bladder distension during and after PAE and the increase in urethral stricture and bladder outlet obstruction due to post-embolization intra-prostatic oedema. Foley catheter insertion during PAE was not necessary as less invasive measures, such as urinating moments prior to entering the Angio suite, a urinal at their disposal during the procedure, and immediate voiding after PAE, were efficient to prevent retention. Furthermore, TRA permits urination in a standing position, moments after the procedure, which facilitates micturition.

Hematuria occurred in 2.6%, which was lower compared to the 5.51, 9, 4.45, and 4.38% previously reported [7,8,9,10]. As the angiographic review reported the close proximity of the vesical arteries to the PA (at least one type-1 PA per patient), these events may suggest that non-target vesical embolization due to reflux of the embolic agent was possibly the cause of hematuria. Nonetheless, the short duration of this event (24 h in all cases except the one associated with prostatic fragment detachment) may suggest the bleeding of prostate tissue necrosis during its healing process following PAE.

One case of 24 h-lasting hematuria and no micturition symptom followed coil protection of an AIVA or a vesical anastomosis, which suggests the safety of this technique for bladder viability and confirms the findings of a previous report in the literature [15]. On the other hand, the absence of coil protection of a vesical artery or anastomose in proximity to the prostatic artery may lead to bladder ischemia: a previous case report reported focal bladder necrosis following non-target vesical embolization during PAE using 100–300 μm microparticles, that was successfully treated by Foley catheter placement for several weeks [23].

There was a 0.6% rate of rectorrhagia, which was lower than the 4.8, 3.9, 3.02, and 3.02% previously reported [7,8,9,10]. The hypothesis for rectal bleeding following PAE is non-target embolization and ischemic ulceration of the rectum. This was previously described in a case report for PAE using 100–300 μm particles, where coil protection was not performed, with the spontaneous resolution of ulcerations within 5 days [24]. In this study, both cases of rectorrhagia did not present arterial anatomy at risk of non-target rectal embolization. As the amount and duration of bleeding were negligible, the traces of blood in stool that were reported by the patients may instead be attributed to hemorrhoid hemorrhage provoked by constipation. Furthermore, there were interestingly 7/12 (58%) cases of pattern C2 PA where the accessory rectal artery could not be coil-protected and was therefore fully embolized with no post-operative complication. This may suggest that the embolization of pattern C2 PAs using 300–500 μm particles does not necessarily require coil protection. This supposition could have been stronger in case of a control group purposely not performing coil protection, but the case report mentioned regarding the above [24] made the safety of such a design questionable.

This hypothesis will need further comparative studies for confirmation, but the safety profile of future study designs must be considered. As the size of the microparticles used may play a role in the occurrence of adverse events [23,24], the choice of particle size of the type of embolic should be carefully made, and choosing to use 300–500 μm particles may be the best option for future randomized trials. In our opinion, the choice of a control group performing the abstention of coil protection may be too risky, and coil protection should be compared to the balloon occlusion micro-catheter technique. Concerning small rectal arteries arising from the PA (pattern C2), the results of this study may suggest another safety profile other than vesical, penile, or large rectal extra-prostatic supplies in case of non-target embolization using 300–500 μm particles.

There was no sign of rectal ischemia following coil protection, which confirmed the results of the previous reports [14,15].

Balanitis was observed in 0.6%, which is comparable to the 0.6, 0.3, and 0.7% reported [8,9,10]. This event occurred in a particular anatomical pattern previously described. Penile adverse events, secondary to non-target microparticle embolization, are more likely to occur in situations where the PA is close to penile vessels, such as pattern B PA, accessory distal pudendal PA, or type-4 PA, and, despite cautious microparticle delivery, can lead to ischemic balanitis or, in the worst case, necrosis of the glans penis [25]. The findings in this study may suggest cautious particle delivery when embolizing the PA in a situation of distal pudendal PA: reflux during embolization may be at a higher risk of adverse events and should be avoided. Coil protection of pattern B PA, when possible, was safe in this study, as previously reported [14,15]. When coil protection is precluded, a balloon occlusion micro catheter may be of use [21] to prevent reflux.

UTIs were found in 0.6% in this cohort, only in patients with a Foley catheter in place because of an AUR, which was lower compared to the 3.1, 2.7, and 3.32% described in the literature [7,9,10]. No distinction was made in previous reports between patients with a Foley catheter or catheter-free. In several studies assessed in meta-analyses, Foley catheter insertion during PAE was performed to facilitate the procedure [4,26,27], which was not the case for patients from this cohort. The findings in this study suggest that the insertion and/or presence of Foley catheter may increase the risk of UTIs.

There was one case of a transient worsening of erectile dysfunction (ED) following the coil protection of a prostato-penile artery in a situation of ipsilateral IPA occlusion. De novo ED following PAE is rare and has been reported [5], but never occurred after coil protection [14,15,28,29,30].

We believe that coil protection should be avoided in situations of poor IPA vasculature. IPA revascularization may also be considered [31] to permit safe bilateral PAE in a two-step process.

There was one case of the detachment of prostate fragments three months after PAE. This rare event was reported in three case reports and a retrospective report in a total of 8 patients and occurred at 2 to 10 weeks: spontaneous tissue elimination of fragment(s) ranging from a 10 to 15 mm diameter and up to 60 mm long was reported after PAE using 250 μm (1 case) [32] or 100–300 ± 300–500 μm microparticles (3 cases) [33], and 4 cases of detachment following 250 μm particles PAE [34,35] required cystoscopic removal of multiple fragments. In the latter (3/48 patients, 6.3%), predictors for the detachment of prostatic tissue were proposed, such as indwelling catheter, high central gland index, and inflammation. Detachment seems to be correlated with the use of small size embolics <300 μm. Torres et al. reported in a randomized study no clinical benefit of 100–300 μm compared to 300–500 μm trisacryl microparticles, and more frequent minor adverse events [17].

Hematospermia was observed in 2.6%, compared to the 3.63, 4.09, and 5.2 reported [7,8,9]. Occurrence may vary from a study to another as its identification relies on sexual activity. Previous reports link its occurrence to the non-target embolization of seminal vesicles [2]. Ischemia of the seminal vesicles following PAE was described [36,37], but as all control MRI in this study did not show morphological/enhancement abnormalities of the SV in patients manifesting hematospermia, bleeding from necrotic prostatic tissue during ejaculation may not be excluded. This is why it was decided in this report to consider hematospermia as a post-embolization symptom rather than an adverse event.

No case of anejaculation was reported. There is, to our knowledge, no report of anejaculation following PAE using 300–500 μm trisacryl microparticles. Anejaculation following PAE ranges in the radiologic literature from 0 to 2.3% [7,8,9,10]. The reasons are unclear. A recent urological report of PAE using 250–400 μm polyzene microparticles [38] reported an unexpected 4/25 (16%) rate of anejaculation and 40% of decreased ejaculation volume assessed by a 4-item sexual questionnaire 3 months after PAE. Among patients facing anejaculation, 3/4 had undergone endoscopic enucleation of the prostate following symptomatic detachment of prostatic tissue after PAE. Furthermore, longer follow-up (>3 months) on ejaculation was not available.

### 3.2. Access-Site Adverse Events

Complications concerning TRA during PAE in this study were comparable to the previous reports on TRA during PAE: there [14,15] were 10% and 3.3% rates of hematoma and radial artery occlusion. Bhatia et al. [39] and Isaacson et al. [40] described 9 and 11% of hematoma and no radial artery occlusion; the number of patients (32 and 19) was lower.

### 3.3. Clinical Success

There was 86.2% of clinical success following PAE, and baseline characteristics evolved favorably within one month with results comparable to what was previously described in the literature [7,8,9,10]. As efficacy was not the topic of this study, and as clinical success was previously shown to be similar between patients undergoing PAE using coil/gelatin protection and the basic technique [14], clinical success was not compared in the population of this study benefiting from coil protection with a control group.

### 3.4. Follow-Up

There was a 12.2% (38/311) rate of loss to follow-up at one month, and 84.9% and 85.2% (264/311 and 265/311) at 6 and 12 months. The amount of missing data after the one-month follow-up needs to be considered. Still, as all but one adverse events were monitored at one month because they occurred during the first days following PAE and had, except for a few rare events, resolved within the first month prior to the one-month follow-up visit, the loss of follow-up may not have a significant impact on estimating the overall occurrence of adverse events and only limits the results on mid-term functional outcomes and clinical success. Furthermore, patients manifesting adverse events lasting longer than one month or occurring between months 1 and 6, such as transient decreased ejaculate volume, prostate fragment detachment, or radial artery occlusion, were all observed in the clinic at 6- and 12-month visits.

This report has some limitations, including its retrospective nature, the limited number of studied patients, the loss of follow-up at mid-term visits, and the short period of follow-up time. Additionally, this study lacks control groups, especially concerning measures used for the protection of extra-prostatic arterial supplies. Most adverse events were retrospectively monitored and based on patient testimony only, and were not confirmed by clinical examination or assessed by urinary/blood tests or imaging at the time of occurrence. This may have led to the over- or underestimation of their occurrence.

## 4. Conclusions

This study showed that outpatient PAE using 300–500 μm calibrated microparticles with improved anatomical knowledge and techniques can lead to fewer and lower rates of minor embolization-related adverse events than previously reported. These findings demand confirmation by complementary studies. The extensive use of coil protection may be questioned, and its comparison to the balloon occlusion microcatheter or other innovative techniques in randomized designs is necessary to assess its utility in different situations of an acknowledged risk of non-target embolization. Furthermore, there is a need to identify predictors for rare adverse events.

## Figures and Tables

**Figure 1 jpm-12-01261-f001:**
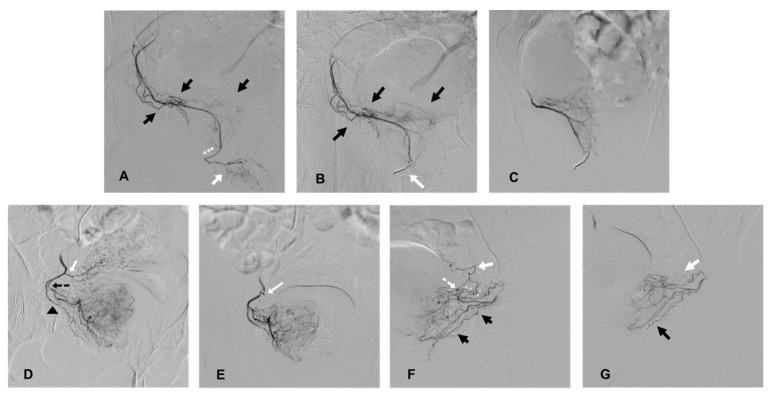
Cases of coil protection of extra-prostatic supplies during PAE and prior to microparticle delivery for safe embolization. (**A**–**C**) present a case of coil protection of a pattern B prostatic artery (PA). (**A**): selective angiography of the right PA on an ipsilateral oblique view, originating from a right accessory internal pudendal artery (APA). Penile arteries are visible at the end of the APA (white arrow) and distally to the prostatic arterial branches (black arrows); the penile bed should be protected from microparticle non-target prostatic embolization (the elective location of occlusion is marked with white asterisks). (**B**): repeat angiography on ipsilateral oblique view, prior to microparticle delivery, and after a 2 and 3 mm diameter detachable microcoil insertion (white arrow). Penile supply is occluded (penile arteries are no longer opacified) and prostatic vessels are still patent (black arrows). (**C**): repeat angiography on anteroposterior (AP) view prior to prostatic embolization for confirmation of a full uptake of the right hemi prostate. Penile supply is still occluded. (**D**,**E**) present a case of occlusion of an accessory inferior vesical artery (AIVA). (**D**): selective angiography of the right PA on ipsilateral oblique view. The tip of the microcatheter is inserted in the medial branch of the PA (marked by a black, dotted arrow) and the lateral prostatic branch is marked by a black arrow head. Early reflux is observed in an ipsilateral AIVA (white arrow) originating from the PA, confirming the risk of non-target embolization. (**E**): repeat angiography on AP view prior to PAE and after the insertion of a 2 mm detachable coil in the AIVA (white arrow). The vesical supply is no longer visible and there is a full uptake of the right hemi prostate. (**F**,**G**) present a case of occlusion of rectal and vesical supplies. (**F**): selective angiography on ipsilateral oblique view of a left prostatic artery, which carries a common trunk with a rectal artery (black arrows), described as pattern C1. There is an associated anastomosis (arterial loop marked by a white, dotted arrow) between the PA and left inferior vesical artery (IVA, white arrow), which needs to be occluded prior to microparticle delivery (elective location marked by white asterisks). (**G**): repeat angiography on oblique view prior to PAE and after the insertion of 2 mm detachable micro coils in the anastomosis to the IVA (white arrow) and in the rectal artery (black arrow), confirming the occlusion of vesical and rectal supplies.

**Figure 2 jpm-12-01261-f002:**
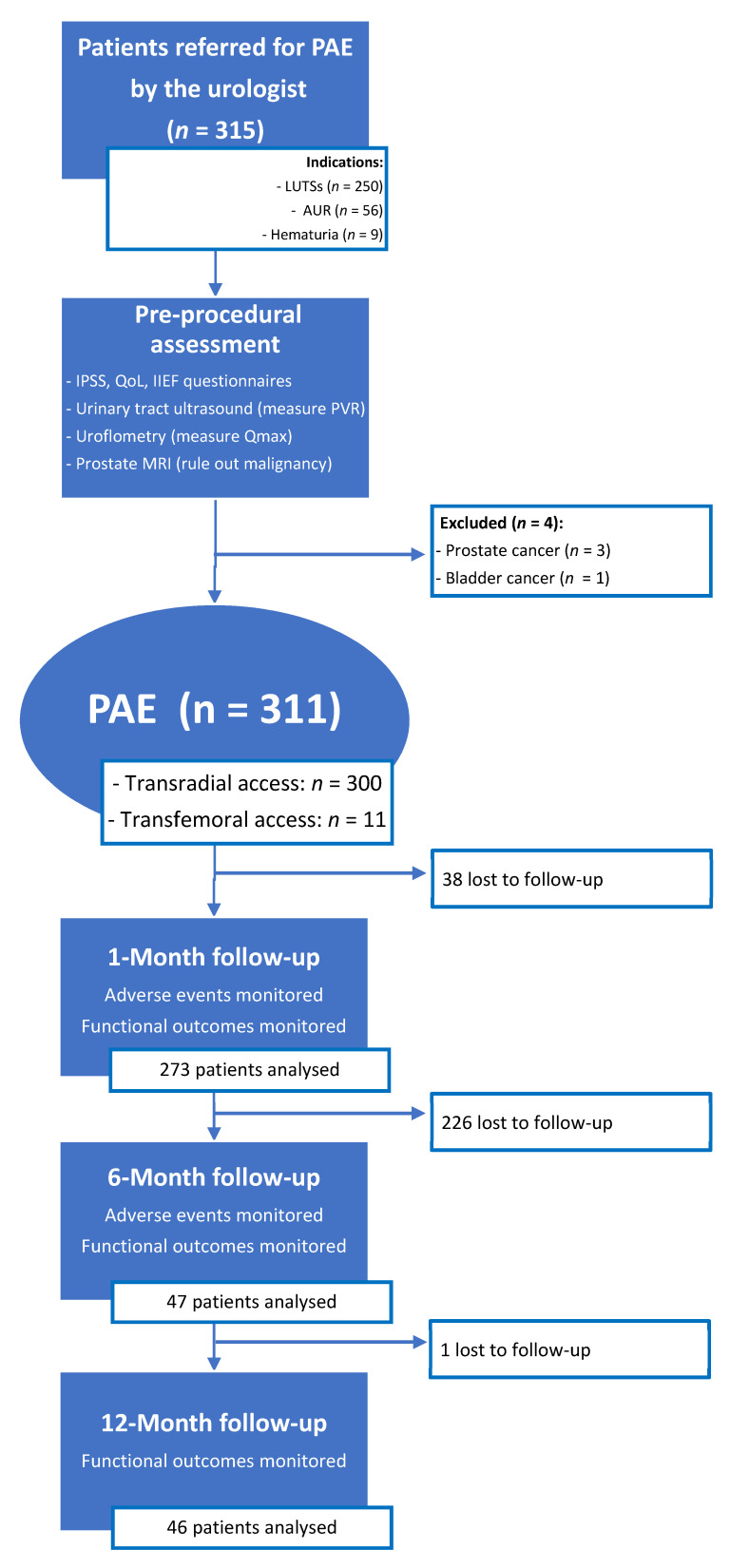
Patient flowchart. PAE: prostatic artery embolization; LUTSs: lower urinary tract symptoms; AUR: acute urinary retention; IPSS: international prostatic symptoms score; QoL: quality of life; IIEF: international index of erectile function; PVR: post-voiding residue; Qmax: maximum urinary flow.

**Figure 3 jpm-12-01261-f003:**
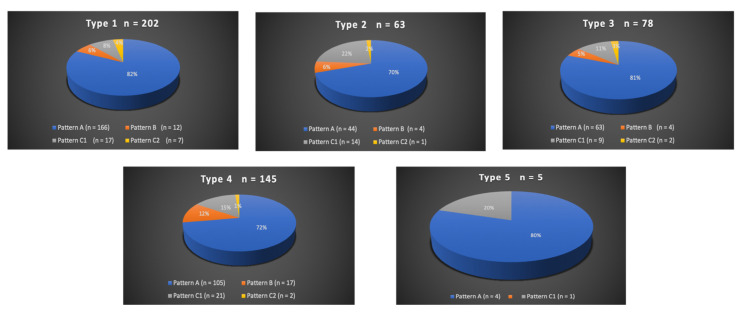
Distribution in study cohort of the origins of the solitary prostatic arteries according to the different patterns. Types 1 to 5 represent the possible origins of the prostatic artery (PA), according to the Assis classification. The values are presented as a number, *n*. Patterns A, B, and C1 and C2 correspond to the different intra/extra-prostatic supplies of the prostatic artery in case of a solitary PA (one artery per side, *n* = 493/622 (79.3%) in this study), according to the Amouyal classification. The values are presented as a number (*n*) and %.

**Figure 4 jpm-12-01261-f004:**
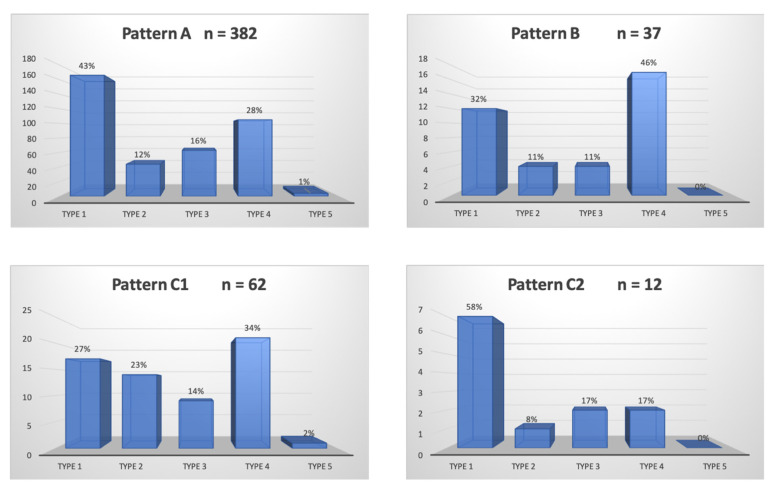
Distribution in study cohort of the patterns of the solitary prostatic arteries according to the different origins. Patterns A, B, and C1 and C2 correspond to the different intra/extra-prostatic supplies of the prostatic artery in the case of a solitary PA (one artery per side, *n* = 493/622 (79.3%) in this study), according to the Amouyal classification. The values are presented as number, *n*. Types 1 to 5 represent the possible origins of the PA, according to the Assis classification. The values are presented as a percentage, %.

**Table 1 jpm-12-01261-t001:** Baseline characteristics of the study cohort.

Variable	Study Cohort (*n* = 311)
Age, yrs	68.7 ± 9.8 (47–102)
Height, cm	176.4 ± 6.5 (158–192)
Radial artery diameter at puncture site, mm	2.5 ± 0.3 (1.7–3.6)
Indication of PAE	
Bothersome LUTSs	246 (79.1)
Urinary retention	56 (18)
Macroscopic hematuria	9 (2.9)
IPSS	18.9 ± 7 (4–35)
QoL score	6 ± 1.1 (2–7)
IIEF-15	44.8 ± 19.5 (4–77)
Prostate volume, mL	91.9 ± 47.3 (22–360)
Maximum urinary flow, mL/s	8.1 ± 5 (2.4–31)
Post-voiding residue, mL	96.6 ± 122.3 (0–810)
Total PSA, ng/mL	6.4 ± 5.6 (0.3–28)

Note: Values are presented as mean ± SD (range) or as number, n (%). PAE: prostatic artery embolization; LUTSs: lower urinary tract symptoms; IPSS: international prostatic symptoms score; QoL: quality of life (range: 1–7); IIEF: international index of erectile function; PSA: prostatic specific antigen.

**Table 2 jpm-12-01261-t002:** Procedure characteristics of the study cohort.

Variable	Study Cohort (*n* = 311)
Transfemoral access	11 (3.6)
Transradial access	300 (96.4)
Unilateral embolization	6 (1.9)
Procedure time, min	96.5 ± 27.4 (45–195)
Fluoroscopy time, min	36.3 ± 15.7 (11–97)
DAP, μGy.m^2^	16,408.3 ± 12,078.9 (2959–81,608)
Radiation skin entry, mGy	1585.7 ± 1115.7 (238–5958)
Mean time to discharge after completion of procedure, min	80.3 ± 7.1 (75–240)
Angiographic review and 3D-angriographic guidance	
Mapping of PA	6 (1.9)
Rule out extra-prostatic supply	4 (1.3)
Coil protection of extra-prostatic supply from prostatic artery	
Vesical accessory artery	23 (7.4)
Prostato-penile anastomose (pattern B)	30 (9.6)
Middle rectal artery from prostato-rectal artery (pattern C1)	20 (6.4)
Accessory rectal artery from prostato-rectal artery (pattern C2)	5 (1.6)
	**Hemi-Pelvis (*n* = 622)**
Solitary prostatic artery per side	493 (79.3)
Multiple prostatic arteries par side	129 (20.7)

Note: Values are presented as mean ± SD (range) or as number, *n* (%). Min: minute; DAP: dose-area product; Gy: gray; PA: prostatic artery; patterns B, C1, and C2 refer to the classification proposed by Amouyal et al.

**Table 3 jpm-12-01261-t003:** Duration of post-embolization syndrome according to prostate volume.

Variable	Mean Duration, Days					
	PV < 40	PV (40–50)	PV (50–80)	PV (80–100)	PV > 100	Overall PV
Mild fever	0	1.4 ± 3.1 (0–7)	0.1 ± 0.3 (0–1)	0.5 ± 1 (0–2)	0.5 ± 1.5 (0–5)	0.4 ± 1.4 (0–7)
Urethral burning	1 ± 1.4 (0–2)	4.2 ± 5.8 (0–11)	3.7 ± 3.8 (0–10)	3.8 ± 3.3 (0–10)	5.1 ± 4.1 (0–12)	4 ± 3.9 (0–12)
Pollakiuria	0	2 ± 3.9 (0–9)	3.8 ± 3.2 (0–10)	4 ± 2.3 (2–9)	6.2 ± 3.1 (3–12)	4.2 ± 3.4 (0–12)
Constipation	0	1 ± 1.4 (0–3)	0.8 ± 1.1 (0–3)	0.3 ± 2.8 (0–2)	1.2 ± 1.6 (0–3)	0.7 ± 1.2 (0–4)
Pelvic pain	1 ± 1.4 (0–2)	6.4 ± 6.3 (0–15)	2.8 ± 5.3 (0–20)	2.8 ± 3.2 (0–9)	2.6 ± 3.8 (0–12)	3.1 ± 4.5 (0–20)
Anal burning/pain	0	3 ± 6.7 (0–15)	2 ± 4.3 (0–15)	0.04 ± 0.8 (0–2)	1.2 ± 2.8 (0–10)	1.5 ± 3.7 (0–15)
Hematospermia	0	7 ± 12 (0–28)	12.4 ± 25.6 (0–90)	4 ± 6 (0–16)	0	6.5 ± 17.2 (0–90)

Note: Values are represented as mean ± SD (range) or as number, *n* (%). PV: prostatic volume (mL) represented as range.

**Table 4 jpm-12-01261-t004:** Embolization-related adverse events.

Variable	Study Cohort (*n* = 311)
Acute urinary retention	0
Foley catheter-related urinary tract infection	2 (0.6)
Urinary tract infection (catheter-free)	0
Hematuria	8 (2.6)
Rectorrhagia	2 (0.6)
Balanitis	2 (0.6)
Detachment of prostatic fragment	1 (0.3)
Worsening of erectile dysfunction	1 (0.3)
Transient ejaculate volume decrease	5 (1.7)
Anejaculation	0
Bladder ischemia	0
Rectal ischemia	0
Penile glans necrotic ulcer	0

Note: Values are presented as mean ± SD (range) or as a number, *n* (%).

**Table 5 jpm-12-01261-t005:** Access-site adverse events.

Variable	TRA (*n* = 300)	TFA (*n* = 11)
Stroke	0	0
Hand pain	0	0
Groin pain	⎯	2 (18)
Hematoma	30 (10)	1 (9)
Pseudo-aneurysm at puncture site	0	0
Thrombosed pseudo-aneurysm at puncture site	2 (0.7)	0
Arteritis	3 (1)	0
Radial artery occlusion	10 (3.3)	⎯

Note: Values are represented as mean ± SD (range) or as a number, *n* (%). TRA: transradial access; TFA: transfemoral access.

**Table 6 jpm-12-01261-t006:** Baseline and follow-up characteristics in study cohort.

Variable	Baseline	1 Month
IPSS	18.9 ± 7 (4–35)	8.7 ± 7.2 (0–35) (*p* < 0.001)
QoL score	6 ± 1.1 (2–7)	3 ± 1.7 (1–7) (*p* < 0.001)
IIEF-15	44.8 ± 19.5 (4–77)	47.6 ± 19.1 (5–72) (*p* = 0.83)
Prostate volume, mL	91.9 ± 47.3 (22–360)	69.4 ± 32 (20–190) (*p* < 0.001)
Maximum urinary flow, mL/s	8.1 ± 5 (2.4–31)	13.4 ± 6.1 (3–32) (*p* < 0.001)
Post-voiding residue, mL	96.6 ± 122.3 (0–810)	39.3 ± 59 (0–270) (*p* < 0.001)
Total PSA, ng/mL	6.4 ± 5.6 (0.3–28)	4 ± 3.1 (0.4–15) (*p* < 0.001)

Note: Values are presented as mean ± SD (range) (*p*-value). Comparison of the data is conducted from its baseline value. *p*-values were obtained using Student’s paired t-test. A *p*-value < 0.05 represents a significant difference. IPSS: international prostatic symptoms score; QoL: quality of life (range: 1–7); IIEF: international index of erectile function; PSA: prostatic specific antigen; mL: milliliter; mL/s: milliliter/second.

## Data Availability

Data containing patient characteristics prior to and after the intervention, in addition to procedure characteristic, are available and can be found in the PACS and RIS of Hospital privé Geoffroy Saint-Hilaire and Hospital Saint-Louis, where the interventions occurred.

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
