# Peer review of "Safety Profile of Ambulatory Prostatic Artery Embolization after a Significant Learning Curve: Update on Adverse Events"

_jpm, 2022, doi:10.3390/jpm12081261_

Round 1
Reviewer 1 Report
Nowadays, prostatic artery embolization (PAE) has been included as a therapeutic option for the treatment of LUTS due to prostatic enlargement. This minimally invasive technique can be performed as a daily procedure under local anaesthesia. PEA has been shown to be as or slightly less effective than transurethral resection of the prostate (TURP) for men with moderate to severe LUTS. However, further studies with medium- and long-term follow-up are needed to evaluate the effectiveness of the procedure and its safety.
One limitation to the spread of this minimally invasive approach is the need for an interventional radiologist with specific training and experience in PAE. Several factors such as atherosclerosis, excessive tortuosity of the arterial supply and the presence of arterial collaterals vasa are anatomical obstacles for this minimally invasive procedure.
In this retrospective study, the authors evaluated the short-term adverse events following PAE performed after a significant learning curve and concluded that experience in arterial anatomy and coil protection may play a role in safety.
This is an interesting and topical study. The manuscript is well structured, but requires some major revisions.
- There are some punctuation and spacing errors in the text;
- Which criteria were used to include or exclude patients from the study(comorbidity, performance status, IPSS, prostate volume, maximum or average flow values, etc)? This must be report in Materials and Methods section;
- The functional results of PEA described in the literature should be discussed in the manuscript and compared to the results of TURP (doi: 10.1016/j.eururo.2021.02.008) and new surgical techniques for the treatment of Benign Prostatic Hyperplasia (doi: 10.3390/app11062467)
- In the manuscript, the authors analysed the complications of PEA. However, it would also be useful to briefly describe the functional outcomes and patient satisfaction in the cohort analysed.
- Some limitations of this study were listed in the manuscript. It would be useful to discuss these in more detail.
Author Response
Response to reviewer 1
Nowadays, prostatic artery embolization (PAE) has been included as a therapeutic option for the treatment of LUTS due to prostatic enlargement. This minimally invasive technique can be performed as a daily procedure under local anaesthesia. PEA has been shown to be as or slightly less effective than transurethral resection of the prostate (TURP) for men with moderate to severe LUTS. However, further studies with medium- and long-term follow-up are needed to evaluate the effectiveness of the procedure and its safety.
One limitation to the spread of this minimally invasive approach is the need for an interventional radiologist with specific training and experience in PAE. Several factors such as atherosclerosis, excessive tortuosity of the arterial supply and the presence of arterial collaterals vasa are anatomical obstacles for this minimally invasive procedure.
In this retrospective study, the authors evaluated the short-term adverse events following PAE performed after a significant learning curve and concluded that experience in arterial anatomy and coil protection may play a role in safety.
This is an interesting and topical study. The manuscript is well structured, but requires some major revisions.
- There are some punctuation and spacing errors in the text;
Thank you. We corrected every error that were found. Please notify if any more should remain.
- Which criteria were used to include or exclude patients from the study (comorbidity, performance status, IPSS, prostate volume, maximum or average flow values, etc)? This must be report in Materials and Methods section;
We described more precisely inclusion criteria and added exclusion criteria.
As described in the added flow-chart suggested by another reviewer, 4 patients referred for PAE did not benefited the intervention, because of malignancy.
- The functional results of PEA described in the literature should be discussed in the manuscript and compared to the results of TURP (doi: 10.1016/j.eururo.2021.02.008) and new surgical techniques for the treatment of Benign Prostatic Hyperplasia (doi: 10.3390/app11062467)
This report focused on safety of performance of PAE. The aim of this study was to report that achieving improvement in anatomical knowledge and technics could lower the incidence and the number of adverse events after PAE, compared to what is, as of now, available in literature.
This study was not focusing on functional outcomes or efficacy of PAE. Comparison between PAE and TURP was already reported in randomized designs, in 4 RCT (references 3 to 6 in the manuscript) and countless reviews/reports.
We don’t really understand why adding a comparison to surgical treatments in the manuscript would bring additional information on the primary outcome.
- In the manuscript, the authors analysed the complications of PEA. However, it would also be useful to briefly describe the functional outcomes and patient satisfaction in the cohort analysed.
We added table 6 and a paragraphe in the manuscript (end of the results section) accordingly.
- Some limitations of this study were listed in the manuscript. It would be useful to discuss these in more detail.
We modified the discussion accordingly.

Reviewer 2 Report
the authors reported their experience on the outpatient prostatic artery embolisation after a significant learning curve with a particular focus on the safety profile. the research was well conducted and well presented, the conclusions were based on reliable results.
I would suggest to add the word "profile" after "safety" in the title.
please, specify the acronym DAP in abstract section.
the materials and methods section lacks the sample size calculation, useful to assess the statistical power of the study.
please, add a flow-chart for included / excluded patients.
I would suggest to focus on a better categorisation of the adverse events, eventually by referring to international grading score/scale systems in order to make results internationally or at least widely considerable and suitable of evaluation / interpretation.
Author Response
Response to reviewer 2
The authors reported their experience on the outpatient prostatic artery embolisation after a significant learning curve with a particular focus on the safety profile. the research was well conducted and well presented, the conclusions were based on reliable results.
I would suggest to add the word "profile" after "safety" in the title.
Was added, thank you for this suggestion.
please, specify the acronym DAP in abstract section.
Was performed, thank you.
the materials and methods section lacks the sample size calculation, useful to assess the statistical power of the study.
As the design of this study was an observational retrospective assessment with no control group and not a prospective randomized study, there was no sample size calculation.
Please, add a flow-chart for included / excluded patients.
This was added to the manuscript. We also added a sentence accordingly in the results and discussion section, according to the lost to follow-up.
I would suggest to focus on a better categorisation of the adverse events, eventually by referring to international grading score/scale systems in order to make results internationally or at least widely considerable and suitable of evaluation / interpretation.
The adverse events were classified in this study according to the Society of Interventional Radiology, which gathers experience of experts in interventional radiology. In overall reports on PAE (and more generally on embolization), this classification was mostly used:
Malling et al (reference 10 In the manuscript, Eur Radiol. 2019;29(1):287-98) reported in a meta-analysis of 13 studies reporting clinical results after PAE that : “SIR classification [18] of complications was used by six studies*, Clavien-Dindo [35] or a modification in three studies, National Cancer Institute Common Terminology Criteria for Adverse Events [36] in two studies, and the remaining two studies did not specify the classification”.
* among the 13 studies was the report of Pisco and al on results in 630 patients benefiting PAE, one of the biggest cohort assessed so far in this topic.
As the aim of this study was to show a decreased rate of adverse events after advanced technics and knowledge in the performance of PAE compared to what was previous reported, this is the main reason why we chose to use the most “standardized” classification in PAE reports to facilitate comparison with these previous studies.
Other classifications could be proposed, but we would be concerned to choose one that is less frequently used in studies on PAE:
- Dindo D, Demartines N, Clavien PA. Classification of surgical complica- tions: a new proposal with evaluation in a cohort of 6336 patients and results of a survey. Ann Surg 2004; 240:205–213.
- Filippiadis DK, Binkert C, Pellerin O, Hoffmann RT, Krajina A, Pereira PL. Cirse quality assurance document and standards for classification of complications: the cirse classification system. Cardiovasc Interv Radiol. 2017;40(8):1141–6.
- Common Terminology Criteria for Adverse Events (CTCAE) Version 4.0 Published: May 28 vJ, 2010).
- US Department of Health and Human Services; National Institutes of Health; National Cancer Institute. Common terminology criteria for adverse events, v4.03. 2010. Available at: https://evs.nci.nih.gov/ftp1/CTCAE/ CTCAE_4.03_2010-06-14_QuickReference_8.5x11.pdf. Accessed April 3, 2017.
Reference 22 has been changed to a more recent version of the SIR guidelines:
doi 10.1016/j.jvir.2017.06.019

Reviewer 3 Report
This manuscript is about clinical experiences with advanced approaches in the treatment of BPH. The authors update on the latest clinical experience in the field. Relevant and important, well written.
-Minor comments:
A limitation of a lack of control groups for several of the methods used should be discussed - it only limits conclusions - make suggestions for future research.
Author Response
Response to reviewer 3
This manuscript is about clinical experiences with advanced approaches in the treatment of BPH. The authors update on the latest clinical experience in the field. Relevant and important, well written.
-Minor comments:
A limitation of a lack of control groups for several of the methods used should be discussed - it only limits conclusions –
Thank you for this remark. Here are our thoughts/concerns on control groups:
- Concerning coil protection and safety of PAE, we agree that control groups would have made the study design more accurate.
- Coil protection vs no measures prior to embolizing:
As several case reports on vesical, rectal, and penile adverse events have been described when coil protection was not performed and are discussed in the manuscript (references 23-25*), a control group performing abstention of protection of the extra-prostatic bed seemed too much at risk and unacceptable from an ethical point of view.
Still, we underlined in the manuscript the fact that some patients did not manifest rectal event after embolization and no prior coil protection of a pattern C2 (small accessory rectal artery). We modified the discussion on this very topic accordingly, thank you.
* we added ref 23, reporting bladder ischemia after PAE to improve discussion around risks of bladder non-target embolization:
- Moschouris H, Stamatiou K, Kornezos I, Kartsouni V, Malagari K. Favorable Outcome of Conservative Management of Extensive Bladder Ischemia Complicating Prostatic Artery Embolization. Cardiovasc Intervent Radiol. 2018;41(1):191-6.
- Coil protection vs balloon occlusion:
This has never been studied and we believe that comparing both techniques may provide valuable data for future practice. This has been added to the discussion and conclusion.
- Concerning coil protection and efficacy of PAE, results have previously been described (ref 14): clinical success was similar in the coil protection group than in the basic PAE group (p = n.s.). Furthermore, efficacy was not the topic of the report, this is why we chose not to compare it to a control group. We added a sentence in the discussion section.
make suggestions for future research.
We added a sentence accordingly in the conclusion, we suppose that the best study in the future should be comparing coil protection to balloon occlusion or any innovative technique that would consist in suppressing the flow in the extra-prostatic supply.
